# Efficacy and safety of using auditory-motor entrainment to improve walking after stroke: a multi-site randomized controlled trial of InTandem™

Louis N. Awad [1,2] ✉, Arun Jayaraman[3], Karen J. Nolan [4,5], Michael D. Lewek[6,7], Paolo Bonato [2], Mark Newman[8], David Putrino [9], Preeti Raghavan[10], Ryan T. Pohlig[11], Brian A. Harris[12], Danielle A. Parker[12] & Sabrina R. Taylor [12]

Walking slowly after stroke reduces health and quality of life. This multi-site, prospective, interventional, 2-arm randomized controlled trial (NCT04121754) evaluated the safety and efficacy of an autonomous neurorehabilitation system (InTandem™) designed to use auditory-motor entrainment to improve post-stroke walking. 87 individuals were randomized to 5-week walking interventions with InTandem or Active Control (i.e., walking without InTandem). The primary endpoints were change in walking speed, measured by the 10-meter walk test pre-vs-post each 5-week intervention, and safety, measured as the frequency of adverse events (AEs). Clinical responder rates were also compared. The trial met its primary endpoints. InTandem was associated with a 2x larger increase in speed (Δ: 0.14 ± 0.03 m/s versus Δ: 0.06 ± 0.02 m/s, F(1,49) = 6.58, p = 0.013), 3x more responders (40% versus 13%, $\chi^2$(1) ≥ 6.47, $p$ = 0.01), and similar safety (both groups experienced the same number of AEs). The auditory-motor intervention autonomously delivered by InTandem is safe and effective in improving walking in the chronic phase of stroke.

Disability after stroke is an important public health issue. The dearth of effective rehabilitation therapies[1] has resulted in stroke remaining a leading cause of long-term disability[2–4]. According to the American Heart Association, the medical-related costs of stroke are estimated to be $50 billion and are projected to more than double by 2035[4]. The personal and societal burdens of stroke necessitate addressing existing care gaps.

A key care gap is the limited efficacy of walking rehabilitation in the chronic phase of stroke recovery. People with chronic stroke impairment identify walking difficulty as a major problem[5] and regard community ambulation "as either essential or very important" to their well-being[6]. Community ambulation is defined as "independent mobility outside the home" and includes confidently navigating uneven terrain, private venues, shopping centers, and other public

[1]Dept. of Physical Therapy, Boston University, Boston, MA, USA. [2]Dept. of PM&R, Harvard Medical School, Spaulding Rehabilitation Hospital, Boston, MA, USA. [3]Dept. of PM&R, Northwestern University, Shirley Ryan AbilityLab, Chicago, IL, USA. [4]Center for Mobility and Rehabilitation Engineering, Kessler Foundation, West Orange, NJ, USA. [5]Dept. of PM&R, Rutgers New Jersey Medical School, Kessler Rehabilitation, Newark, NJ, USA. [6]Dept. of Health Sciences, University of North Carolina at Chapel Hill, Chapel Hill, NC, USA. [7]Division of Physical Therapy, University of North Carolina at Chapel Hill, Chapel Hill, NC, USA. [8]Dept. of PM&R, Carolinas Rehabilitation, Charlotte, NC, USA. [9]Abilities Research Center, Icahn School of Medicine at Mount Sinai, New York, NY, USA. [10]Depts. of PM&R & Neurology, Johns Hopkins University School of Medicine, Baltimore, MD, USA. [11]College of Health Sciences, University of Delaware, Newark, DE, USA. [12]MedRhythms, Inc., Portland, ME, USA. ✉ e-mail: louawad@bu.edu

settings[7]. However, as many as 85% of individuals with chronic stroke impairment have at least some limitation in community ambulation[8]. To navigate community settings successfully, a minimum walking speed of 0.80 m/s[9] is often recommended. Whereas the average walking speed of community-dwelling older adults is 1.3 m/s[10], individuals in the chronic phase of stroke have an average speed of only 0.74 m/s[6]. That is, the average patient likely does not walk fast enough to navigate most community settings.

Slow walking is associated with a high fall risk, increased comorbidities, and reduced quality of life[11–15]. Interventions that are effective in improving the walking speed of people with chronic hemiparesis after stroke are critically needed; however, an additional care gap that exists in parallel with, and contributes to the limited *efficacy* of post-stroke walking rehabilitation is limited *access* to skilled interventions. Disparities in time, childcare, work responsibilities, and access to reliable transportation[16] often prevent community-dwelling individuals with mobility impairment from engaging in the rehabilitation needed for long-term health[9]. Together, care gaps in treatment *efficacy* and *access* necessitate the advancement of interventions that can provide safe and effective walking rehabilitation across care settings, including the home and the community.

We present the InTandem autonomous neurorehabilitation system as a promising solution to address these care gaps. InTandem aims to improve the walking of individuals in the chronic phase of stroke recovery by autonomously applying the principles of rhythmic auditory stimulation[17–28]. The system's primary mechanism of action is auditory-motor entrainment, a neurally-mediated process whereby the timing of motor movements is involuntarily synchronized with the timing of a rhythmic auditory stimulus (e.g., music featuring a strong beat). The involuntary synchronization between movement and auditory rhythm allows InTandem to autonomously provide a progressive intervention by adjusting elements of the auditory rhythm, like tempo and beat salience. Notably, the rhythm-based intervention is individualized to each user's unique gait pattern by the combination of real-time gait sensing with closed-loop music control algorithms, eliminating the need for clinician and user input to safely increase walking speed within and across intervention sessions.

Two proof-of-concept studies conducted with InTandem research prototypes demonstrate its clinical promise[29,30]. The first proof-of-concept study reported clinically meaningful gains in walking speed after one intervention session[29]. Additional sessions resulted in continued improvements in walking speed, with an average increase of 0.12 ± 0.03 m/s observed after three sessions. The second proof-of-concept study reported a 22 ± 10% median reduction in multiple measures of temporal gait asymmetry and a median 9 ± 5% reduction in the energetic cost of walking after one session[30]. These findings are indicative of improvements in the neuromuscular control of walking, which is important given that the gait of individuals post-stroke is not only slow, but also highly asymmetric and effortful[31]—factors associated with reduced physical activity and worse long-term health[32].

Building on this foundational prior work, we conducted a multi-site, prospective, interventional, 2-arm randomized controlled trial to evaluate the safety and efficacy of InTandem relative to a treatment-matched Active Control group. Importantly, though the primary outcome of this study of the autonomous InTandem intervention is an improvement in walking speed, InTandem's algorithm incorporates real-time assessment of gait quality to determine when progression is appropriate. That is to say, InTandem balances walking *safely* alongside walking *faster*; this feature of the intervention is believed to be a critical element to autonomous use of InTandem in free-living settings by individuals at high risk of falling, such as individuals with chronic post-stroke hemiparesis. Safety was evaluated as the frequency of adverse events (AEs). Efficacy was evaluated by testing the hypothesis that, compared to the Active Control group, InTandem would result in a greater increase in walking speed and a greater percentage of responders. We also tested the exploratory hypothesis that the individualized and progressive intervention would result in a faster rate of walking speed improvement over the 5-week intervention period.

## Results

### Participants
A total of 87 individuals were enrolled and randomized. Of these, six withdrew after randomization and before completing their first session. The remaining 81 individuals completed at least one session and were thus to be included in the trial's intent-to-treat analysis. However, of these 81, one statistical and clinical outlier was identified and removed based on pre-planned outlier handling. This participant had a change in 10 mWT speed of 0.79 m/s, which was 3.45 standard deviations beyond the group mean and would be considered non-typical for five weeks of gait rehabilitation[33]. As noted in Methods: COVID-19 Considerations, eight additional participants were administratively removed from the trial after a COVID-19-related investigation into the trial data. Four of these study participants were from the InTandem intervention group, and four were from the Active Control group.

The trial's intent-to-treat analysis thus included 72 participants who completed at least one walking session. These study participants had an average (± standard deviation) time since stroke of 8.1 ± 7.1 years and were 62.3 ± 7.0 years of age. The cohort was 58.3% male, 43.1% Black or African American, and 80.6% Not Hispanic or Latino. Approximately 77% of participants completed some college or beyond. There were no significant demographic differences between groups (Table 1). Of these 72 participants, four randomized to InTandem, and nine randomized to Active Control did not complete the full 15-session intervention schedule (see consort diagram in Fig. 1).

### Session completion and InTandem system reliability
The trial included a total of 1015 completed intervention sessions. Of these 1015 sessions, 984 sessions (96.9%) were fully complete and 31 (3.1%) were terminated early. Of the 31 sessions terminated early, 15 were ≥15 min in duration and thus did not require restart or rescheduling as per the trial protocol. The remaining 16 sessions that were terminated early were <15 min and thus considered incomplete, requiring a session restart or rescheduling. Of these 16 incomplete sessions, 13 were incomplete due to software or system component issues. System-related issues were able to be rapidly resolved, allowing the sessions to be restarted and completed uneventfully on the same day without a need to reschedule. The remaining 3 incomplete sessions were incomplete due to the participant being too tired to continue ($n = 1$) and AEs ($n = 2$); none of the 3 sessions were able to be restarted on the same day. Two of these three incomplete sessions were rescheduled to a later date and completed successfully; the one remaining incomplete session was never rescheduled as the participant was withdrawn from the trial due to unrelated medical reasons.

### Primary endpoint analyses
The trial's primary endpoint was a between-group difference in the change in self-selected comfortable walking speed, as measured using the 10 mWT (i.e., post-intervention 10 mWT speed−pre-intervention 10 mWT speed). InTandem resulted in a larger increase in 10 mWT speed compared to the Active Control intervention ($F(1, 49) = 6.58$, $p = 0.013$). Specifically, InTandem resulted in an average (± standard error) increase in 10 mWT speed of 0.14 ± 0.03 m/s ($p < 0.001$) compared to the Active Control's increase of 0.06 ± 0.02 m/s ($p < 0.001$) (Fig. 2A). Moreover, InTandem resulted in a greater proportion of responders based on the trial's two definitions of being a responder (see Methods: Statistical Analyses). Participants randomized to InTandem were 3.2x more likely to increase their 10 mWT speed beyond the 0.16 m/s Minimal Clinical Important Difference (MCID)

**Table 1 | Baseline demographic and clinical characteristics of participants**

| Descriptive variable | Cohort A InTandem (n = 40) | Cohort B active control (n = 32) | Overall (n = 72) | Sig. (p) |
|---|---|---|---|---|
| Age | | | | 0.168 |
| Mean (SD) | 61.3 (6.5) | 63.6 (7.5) | 62.3 (7.0) | 0.168 |
| Gender | | | | 0.200 |
| Male | 26 (65.0%) | 16 (50.0%) | 42 (58.3%) | |
| Female | 14 (35.0%) | 16 (50.0%) | 30 (41.7%) | |
| Ethnicity | | | | 0 .797 |
| Hispanic or Latino | 5 (12.5%) | 2 (6.3%) | 7 (9.7%) | |
| Not Hispanic or Latino | 31 (77.5%) | 27 (84.4%) | 58 (80.6%) | |
| Unknown | 2 (5.0%) | 2 (6.3%) | 4 (5.6%) | |
| Not reported | 2 (5.0%) | 1 (3.1%) | 3 (4.2%) | |
| Race | | | | |
| American Indian or Alaska Native | 0 (0.0%) | 0 (0.0%) | 0 (0.0%) | |
| Asian | 0 (0.0%) | 1 (3.1%) | 1 (1.4%) | |
| Black or African American | 16 (40.0%) | 15 (46.9%) | 31 (43.1%) | |
| Native Hawaiian or Other Pacific Islander | 0 (0.0%) | 0 (0.0%) | 0 (0.0%) | |
| White | 17 (42.5%) | 14 (43.8%) | 31 (43.1%) | |
| Identified as more than 1 race | 2 (5.0%) | 0 (0.0%) | 2 (2.8%) | |
| Unknown | 2 (5.0%) | 0 (0.0%) | 2 (2.8%) | |
| Not reported | 3 (7.5%) | 2 (6.3%) | 5 (6.9%) | |
| Education level | | | | 0.894 |
| Some high school | 3 (7.5%) | 1 (3.1%) | 4 (5.6%) | |
| High school graduate | 7 (17.5%) | 6 (18.8%) | 13 (18.1%) | |
| Some college | 14 (35.0%) | 11 (34.4%) | 25 (34.7%) | |
| College graduate | 12 (30.0%) | 9 (28.1%) | 21 (29.2%) | |
| Graduate degree | 4 (10.0%) | 5 (15.6%) | 9 (12.5%) | |
| Time since stroke (years) | | | | 0.684 |
| Mean (SD) | 8.4 (6.9) | 7.7 (7.6) | 8.1 (7.1) | |

Independent *t* tests (two-sided) were used to evaluate between-group differences in these baseline demographic and clinical characteristics. Mean values ± standard deviations and *p* values are reported for each analysis.

(Active Control: 4/32, 12.5%; InTandem: 16/40, 40.0%; $\chi^2(1) = 6.70$, $p = 0.01$, Cramer's $V = 0.31$) and 3.7x more likely to *both* increase their 10 mWT speed beyond the 0.16 m/s MCID and have a post-intervention speed above 0.80 m/s (Active Control: 3/32, 9.4%; InTandem: 14/40, 35.0%; $\chi^2(1) = 6.47$, $p = 0.01$, Cramer's $V = 0.30$) (Fig. 2B).

**Exploratory time-course of change analysis**

In addition to measuring 10 mWT speed before and after each intervention period, 10 mWT speed was also measured before and after each treatment session, enabling evaluation of the time-course of changes in walking speed. The average 10 mWT speed measured before and after each session changed by a different rate across the 5-week intervention period for the InTandem group versus the Active Control group (i.e., a significant Session x Treatment interaction, $F(16, 206) = 1.790$, $p = 0.034$). Though both the InTandem and the Active Control groups increased their 10 mWT speed across sessions, individuals who received the InTandem intervention showed a markedly faster rate of walking speed increase across sessions (Fig. 2C).

**Safety: frequency of adverse events**

Seven InTandem users (17.5%) experienced a total of ten AEs and two serious adverse events (SAEs) over the course of the trial (Table 2). One SAE was deemed possibly related to InTandem because it occurred during a walking session; the event resolved within a day and the participant completed the trial on schedule. Similarly, in the Active Control group, seven participants (21.9%) experienced AEs, two of which were considered SAEs. Neither SAE was deemed related to the walking sessions.

There were six reported fall events during the trial: four falls in the Active Control group compared to two falls in the InTandem group. All six falls were non-injurious, occurred outside of trial visits, and were considered "not related" or "unlikely related." There was no observed pattern to the timing of the fall events with respect to the 5-week training period; two falls occurred during study participants' first week of training, and one fall occurred in each of the subsequent weeks of the training period (i.e., one in week 2, one in week 3, etc.).

## Discussion

InTandem is an autonomous neurorehabilitation system that applies the closed-loop control of music to provide an individualized and progressive intervention tailored to each user's unique gait pattern, without requiring input from clinicians or users. The findings of this multi-center randomized clinical trial demonstrate the safety and efficacy of InTandem for individuals with chronic walking impairment after stroke. When taken together with findings of high usability[34], InTandem has potential to address key efficacy and access care gaps.

**Faster walking speeds and rate of recovery**

Autonomous rehabilitation delivered by the InTandem system resulted in significantly faster post-training walking speeds compared to the treatment-matched Active Control. Given the relationship between walking speed and health, function, and quality of life, success in achieving the trial's primary endpoint of a between-group difference in the change in walking speed underscores the potential of the auditory-motor intervention delivered by the InTandem system to improve walking after stroke. Critically, regardless of the responder analysis used, InTandem participants were three times more likely to be responders. The a priori selection of two responder analyses for this study was based on extensive discussions with different stakeholders (i.e., FDA, users, prescribers, and payers), wherein different groups were found to value the two responder analyses differently. In brief, the most common approach to defining a responder is to use an MCID cutoff; however, a key limitation of this approach is that a subject's walking speed change may surpass the MCID but not be sufficient to place their absolute walking speed above clinically meaningful thresholds (e.g., 0.80 m/s is a walking speed threshold that is thought to be the minimum required for community walking). In contrast, a key limitation to defining a responder only as someone who surpasses an absolute walking speed threshold, without regard for the magnitude of their change in speed, is that a modest, non-clinically important change could be sufficient (e.g., a 0.02 m/s change for someone with a baseline speed of 0.79 m/s would raise their absolute walking speed over the 0.80 m/s threshold). For our study, by having the first responder analysis focus on the 0.16 m/s MCID cutoff, the analysis is able to compare well to other papers in the field. And by having our second responder analysis combine both criteria (i.e., a change >0.16 m/s and a post-training speed of >0.80 m/s), we are able to address the limitations to using each alone. Ultimately, though these two responder analyses produced similar results in our study, this may not be true for other intervention studies, and we would encourage others to consider this dual approach in future designs to maximize the scientific reach of their work.

InTandem participants also experienced a markedly faster rate of walking speed improvement throughout the five-week intervention

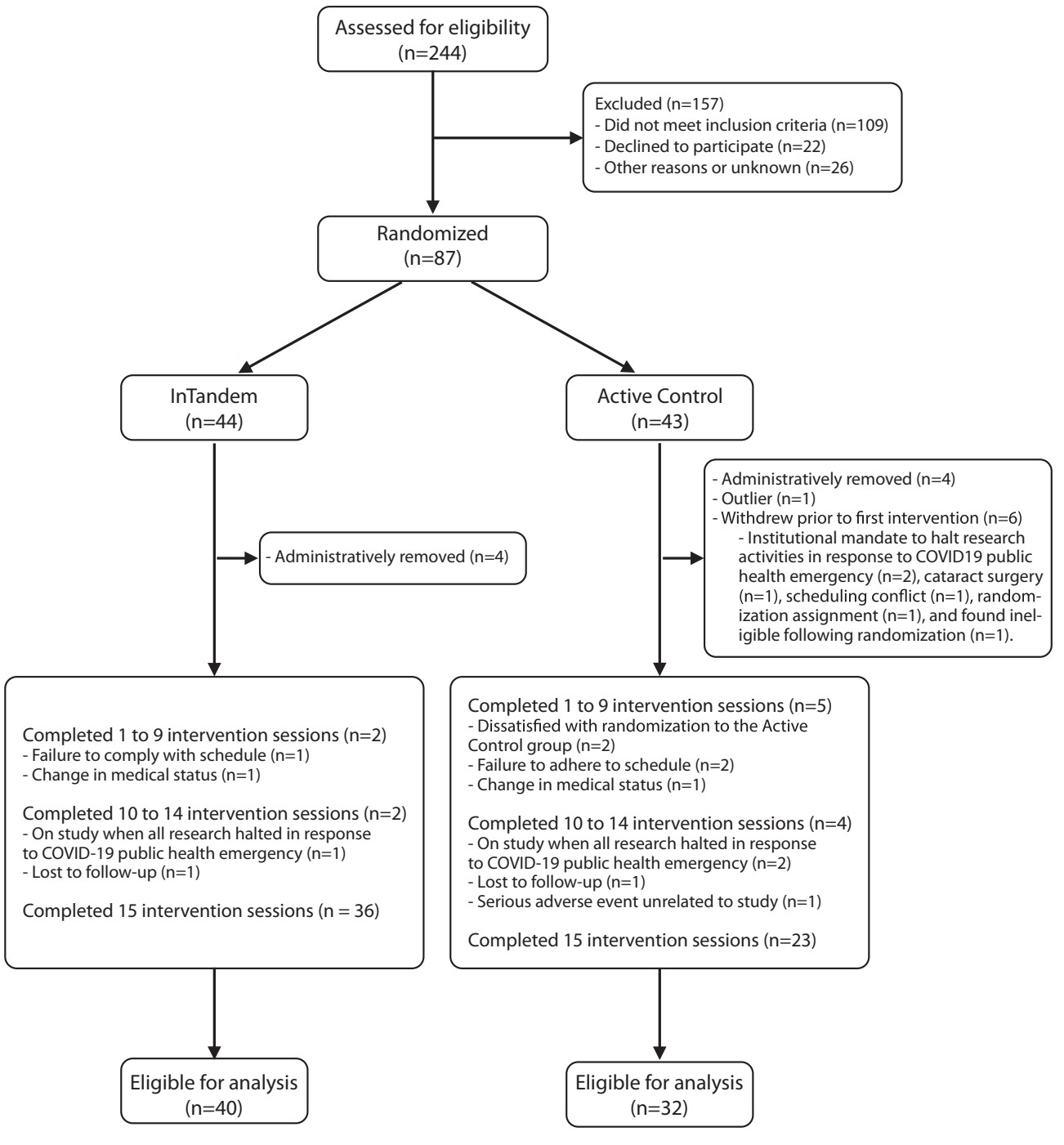

**Fig. 1 | Consort diagram.** Consort diagram of trial population.

period. Whereas the Active Control intervention required 16 trial sessions to produce an average walking speed change of 0.05 m/s, InTandem facilitated this change every 6 trial sessions. The difference in the rate of speed recovery across the intervention groups has two major implications. On the one hand, during the early trial period (i.e., sessions 1 through 8), the effect of the two interventions on walking speed showed a rapid separation. On the other hand, during the latter half of the trial period (i.e., sessions 9 through 16), there was no evidence of a plateau in speed recovery in either group. This suggests that an even longer intervention period may produce even larger gains. Indeed, a recent randomized clinical trial by Boyne et al. tracked changes in the walking speed of individuals with chronic stroke walking impairment that resulted from 12 weeks of high versus moderate-

intensity walking rehabilitation, reporting growing separation between the two intervention arms from 4 to 8 to 12 weeks of training[35]. These findings motivate further study of the effects of longer-term neurorehabilitation with InTandem.

Comparison of the findings of Boyne et al. to our study suggests that InTandem may be more effective than both moderate-intensity aerobic training and high-intensity interval training. Indeed, whereas the moderate-intensity aerobic training group studied by Boyne et al. performed similarly to our study's Active Control group (i.e., both had a 0.06 m/s average increase in 10 mWT speed), when compared to high-intensity interval training, InTandem resulted in larger gains in less time. More specifically, InTandem resulted in a 27% larger increase in 10 mWT speed (i.e., 0.14 m/s vs. 0.11 m/s) in 37% less time (i.e., 5 weeks vs.

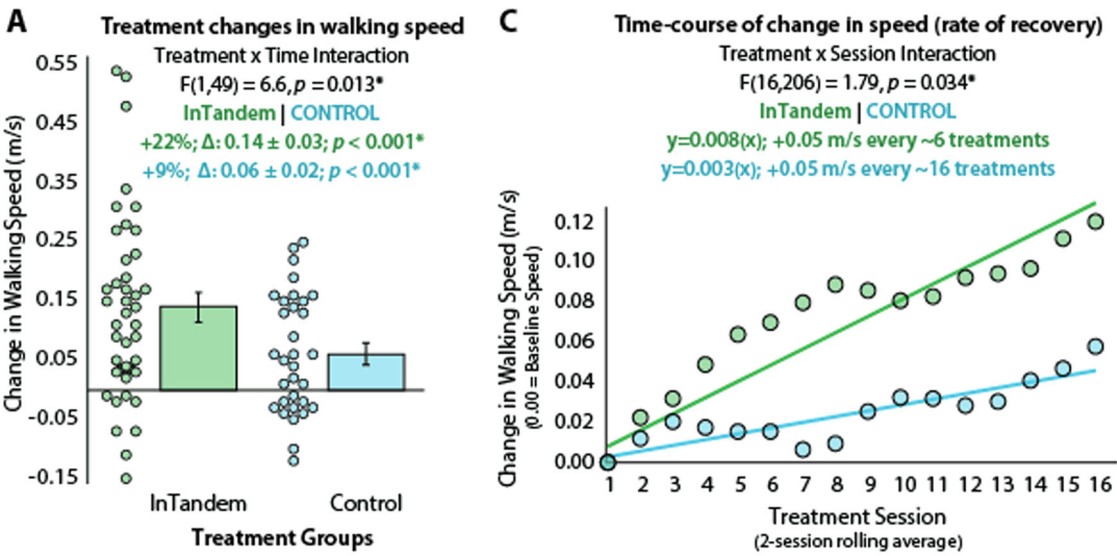

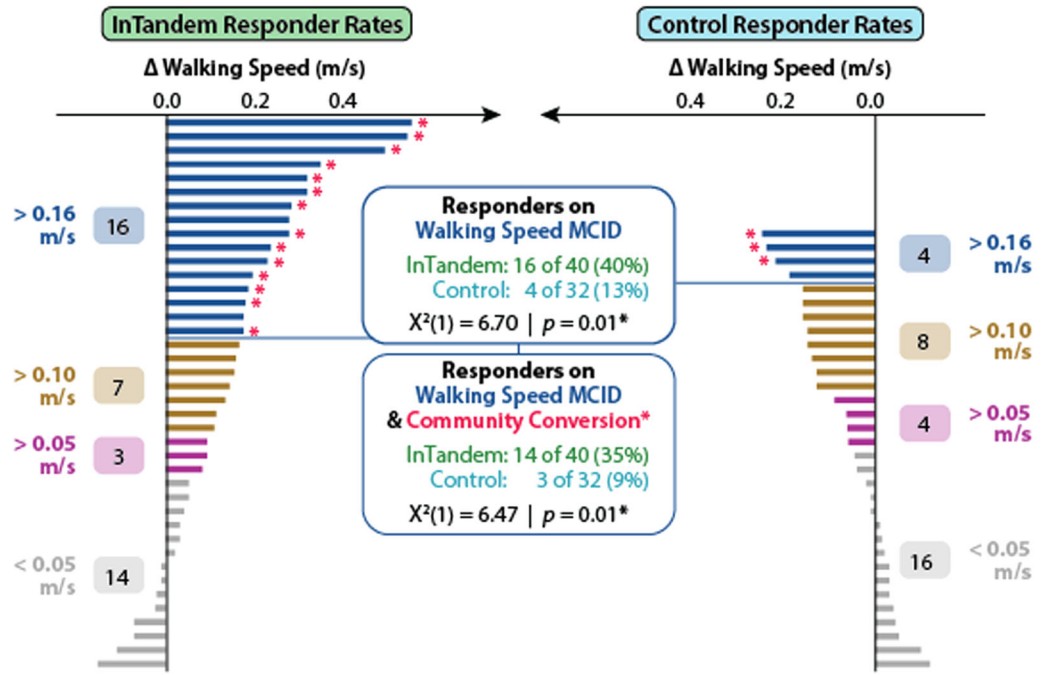

**Fig. 2 | Primary endpoint and responder analyses.** Between-group differences in **A** treatment-related change in walking speed; **B** responder rates—with responder defined as (1) a change in walking speed larger than the 0.16 m/s MCID or (2) a change in walking speed >0.16 m/s MCID *and* a post-training walking speed above 0.80 m/s; and **C** the time course of walking speed improvement. The trial's intent-to-treat analyses included 72 study participants ($n = 40$ in the InTandem group [shown in green] and $n = 32$ in the Active Control group [shown in blue]). **A** reports individual subject walking speed changes and the mean values ± standard error for each treatment group. Shown are the results of a 2 × 2 General Linear Mixed Model (GLMM) used to evaluate between-group differences in the treatment effect (i.e., the treatment × time interaction: $F(1,49) = 6.6$, $p = 0.013$). **B** reports individual

subject data points relative to different clinically meaningful speed change thresholds. Shown are the results of two Chi-Square ($\chi^2$) tests used to evaluate between-group differences in the number of responders (responder analysis 1: $\chi^2(1) = 6.70$, $p = 0.01$; responder analysis 2: $\chi^2(1) = 6.47$, $p = 0.01$). Note: the horizontal axes for the InTandem (left) and Control (right) groups are oriented in opposite directions. **C** reports the results of a 2 × 17 × 2 GLMM used to evaluate between-group differences in the time course of change in walking speed (i.e., the Treatment × Session interaction: $F(16,206) = 1.79$, $p = 0.034$). Each data point is the inter-subject session average within each group. Source data are provided as a Source Data file.

8 weeks). However, it should be noted that Boyne et al. report continued gains in walking speed, up to 0.19 m/s, after 12 weeks of high-intensity training, which we are unable to directly compare against given the shorter intervention duration of our study. Nonetheless, the trajectory of improvement observed in our time-course of change analysis (Fig. 2C) may suggest InTandem's potential to produce larger gains than high-intensity interval training in less time.

## Faster walking, but not at the expense of safety

InTandem was generally safe and well-tolerated, with an equal number of AEs and SAEs as reported for the Active Control group. Moreover, the number of treatment-emergent falls were fewer (i.e., two in the InTandem group versus four in the Active Control group). Critically, it should be noted that the similar number of adverse events was observed despite InTandem users walking markedly faster throughout

**Table 2 | Relatedness and severity of reported adverse events**

| Adverse event relatedness and severity | InTandem | Control |
|---|---|---|
| **Not related** | | |
| Mild | | |
| covid (infected) | 2 | |
| covid vaccine | 1 | |
| fall | 1 | 4 |
| headache | | 1 |
| hypertension | 1 | |
| pain | 2 | |
| tonsillitis | 1 | |
| Moderate | | |
| hospitalization | | 1[a] |
| Severe | | |
| weakness | 1[a] | |
| Life-threatening | | |
| seizures | | 1[a] |
| **Unlikely related** | | |
| Mild | | |
| fall | 1 | |
| **Possibly related** | | |
| Mild | | |
| hypertension | 1 | |
| Severe | | |
| chest pain/diaphoresis/tachycardia | 1[a] | |
| **Probably related** | | |
| Mild | | |
| pain | | 5 |
| Grand total | 12 | 12 |

[a]Denotes ent (SAE)

most of the study period (Fig. 2C). Though these results challenge the common belief that faster walking may be unsafe for individuals post-stroke, to address this reasonable concern, the autonomous InTandem intervention incorporates a multi-tiered decision-making framework to determine when progression of the music's tempo is appropriate (see Methods). The real-time assessment of gait quality included in this decision-making process is considered a key safety feature. By integrating continuous evaluations of gait quality into the system's decision-making processes, future deployment of the InTandem intervention for targeted walking rehabilitation during everyday activities is likely to be safe and well-tolerated by individuals with chronic gait impairment after stroke.

In summary, the InTandem neurorehabilitation system is safe and effective in improving walking after stroke. Applying the closed-loop control of music to autonomously individualize and progress post-stroke walking rehabilitation has potential to address the unmet efficacy and access needs that currently limit recovery. While further study is warranted to better understand the feasibility and rehabilitative potential of using InTandem in home settings, the ideal length of treatment, the durability of effect, and healthcare resource utilization impact, the findings of this study support advancement of InTandem as a promising treatment option for individuals in the chronic phase of stroke.

## Limitations
Several limitations to the trial must be acknowledged. First, the trial's inclusion criteria limited the intervention to individuals with chronic post-stroke hemiparesis with walking speeds between 0.50 m/s and 0.80 m/s. Though this is the subgroup of people post-stroke we predicted would most benefit from this intervention, generalizability to other subgroups post-stroke, as well as other patient populations, is unknown. Second, the trial's five-week intervention period was relatively short, and it is unknown if the observed rate of change would continue with longer treatment periods. Third, there was no follow-up period. In traditional rehabilitation RCT designs, follow-up assessments are often necessary to evaluate motor learning (i.e., retention) and the durability of the treatment effect; however, in contrast to traditional designs that evaluate the primary endpoint across three timepoints (i.e., pre-training, post-training, and follow-up), our time-course of change analysis allowed assessment of changes in walking speed across 17 timepoints (i.e., pre-training, each of the 15 training sessions, and post-training), providing evidence for marked motor learning (i.e., retention) session-over-session in the InTandem group that was largely absent in the Active Control group. However, without a follow-up assessment, the durability of InTandem's effects are not known. Fourth, the trial's primary efficacy analysis did not include other important outcome measures, such as gait biomechanics, patient-perceived benefit, self-efficacy, and community walking activity; the full extent to which post-stroke walking can be improved by InTandem is thus not known. Finally, though study participants were blinded to their 10 mWT speeds, the nature of the intervention prevented blinding of group assignment. Future clinical trials are warranted to assess InTandem's generalizability to other subgroups and patient populations, as well as any additional benefit from longer treatment periods, durability of treatment effect, and impact on other important outcome measures, including gait biomechanics. The evaluation of InTandem's effects on post-stroke gait, within and across intervention sessions, is a natural area for future study given that the InTandem system inherently measures gait parameters to individualize and progress the auditory-motor intervention.

Lastly, while the InTandem neurorehabilitation system is designed to provide an individualized and progressive intervention without requiring a clinician to provide real-time input, it is important to acknowledge that InTandem does not replace the need for a clinician prescriber. In practice, a clinician will identify appropriate candidates for the automated auditory-motor intervention and may be required to define patient-specific conditions for its implementation that go beyond the intervention itself—e.g., the use of adjunctive therapies and training aides, the level of supervision and guarding that may be required to walk, physiological parameter limits (e.g., maximum allowable heart rate), or appropriate settings of use.

## Methods
The multi-site, prospective, interventional, 2-arm randomized controlled trial of the safety and efficacy of the InTandem neurorehabilitation system enrolled 87 participants. The trial was fully enrolled, with the first patient entering the study on September 17, 2019 and the final patient enrolled on January 7, 2022. Per regulations 42 CFR 11.24(a) for an applicable clinical trial for which registration information is required to be submitted, the trial was registered on clinicaltrials.gov on October 7, 2019 (NCT04121754). Consistent with the regulation, all applicable registration information was submitted within 21 days of the first subject being enrolled. The brief delay was necessary due to the release of a new revision of the study protocol on September 16, 2019, which required notification to sites and respective IRB submission. The delay does not bias the study results given that only baseline assessments were completed prior to the initial release of the study registration. The trial complied with all relevant ethical regulations and was approved by the Institutional Review Boards for each participating center as follows: Advarra for Carolinas Medical Center, Boston University Charles River Campus IRB (which oversaw activities at both Boston University and Spaulding Rehabilitation Hospital), IRB of the Icahn School of Medicine at Mount Sinai, Johns Hopkins Medicine IRB, Kessler Foundation IRB, Northwestern University IRB, and UNC at Chapel Hill Non-Biomedical IRB. Written

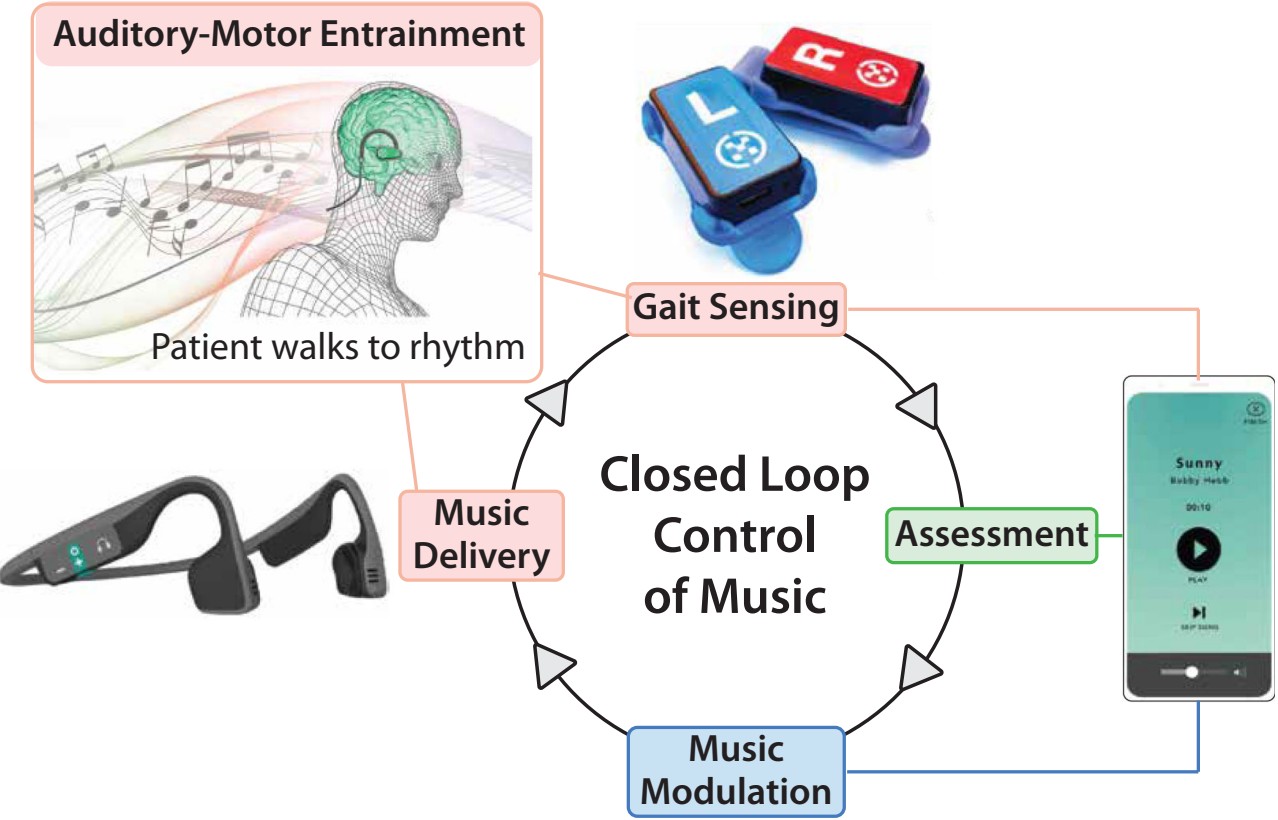

**Fig. 3 | Overview of the InTandem Autonomous Neurorehabilitation System.** The InTandem system combines inertial sensors, a rhythm delivery device, and a locked touchscreen device to provide individualized and progressive walking rehabilitation via the closed-loop control of music. The system's primary mechanism of action is auditory-motor entrainment.

informed consent was secured for all participants. Participants were compensated for their time. All procedures were in accordance with institutional guidelines.

### Inclusion and exclusion criteria
Inclusion criteria for the trial consisted of the following: ≥50 years of age, greater than six months post-stroke, a baseline speed ≥0.50 m/s and <0.80 m/s, gait asymmetry observed by the study investigator, able and willing to consent, no cognitive impairment as evidenced in score ≤1 on question 1b and a 0 on question 1c on the NIH Stroke Scale, and ability to safely participate in protocol-defined sessions of 30-minute duration. Exclusion criteria for the trial consisted of the following: participating in another trial to evaluate an investigational drug at the time of enrollment, previous completion of the trial, unable or unwilling to provide informed consent, unable to participate in protocol-defined sessions without the use of assistive devices (e.g., a cane or walker), a baseline walking speed >0.80 m/s, known history of neurologic (excluding stroke) injury, more than 2 falls in the previous month, active participation in another walking rehabilitation intervention (e.g., physical therapy), use of an external lower limb prosthetic (i.e., artificial limb), hearing impairment that prevents interaction with the InTandem system, self-report of orthopedic surgery in the last year, severe aphasia and/or a speech/language disorder that limits the ability to express needs and comprehend instructions, comorbidities that prevent participation in a rehabilitative walking intervention, or vulnerable populations deemed inappropriate for participation in the trial.

### Data collection overview
Trial data were collected at a baseline screening assessment, during each intervention session, and at a trial closing visit. The baseline screening consisted of administration of NIH Stroke Scale question 1b and 1c and the collection of basic demographic and medical history data. Medical history data were reviewed throughout the trial to identify changes. The 10-meter walk test (10 mWT; comfortable speed) was used to measure the trial's primary endpoint and was conducted at the baseline screening, the trial closing visit, and to start and end each intervention session. AEs were collected at each trial visit, or volunteered by the participant between trial visits, and captured on a standard form. The standard AE form collected a description of the event, the onset and resolution dates (or if the event was ongoing), the severity, management/treatment, outcome, and determination of the relationship to the intervention.

### Primary endpoint data collection
The trial's primary endpoint was a between-group difference in the change in self-selected comfortable walking speed, as measured using the 10 mWT (i.e., post-intervention 10 mWT speed – pre-intervention 10 mWT speed). The 10 mWT is a timed walk test on a 10-meter walkway, with only the middle 6 meters timed. The 10 mWT speed is the average walking speed from three separate 10 mWT trials and is computed in meters per second (m/s). Assistive devices (i.e., cane, crutch, walker, and/or functional electrical stimulation device) were not allowed during 10 mWT assessments; however, lower limb orthoses and braces were permitted if necessary for safety. Participants were blinded to the results of their 10 mWT.

The primary endpoint analysis used a pre-intervention 10 mWT speed collected no more than 14 days before the first session. If scheduling challenges required more than 14 days between the baseline assessment and the first session, the 10 mWT speed measured immediately prior to the start of the first session was used. The post-intervention 10 mWT speed used in the primary endpoint analysis was collected no

more than 4 days after the final session. If scheduling challenges required more than 4 days between the trial closing visit and the final session, the 10 mWT speed measured immediately prior to the final session was used. If participants did not complete the full 5-week intervention schedule or did not return to complete the trial closing visit, the 10 mWT speed used in the intent-to-treat analysis was the 10 mWT speed measured immediately prior to the final session completed.

### Randomization

Eligible participants were randomized to one of two intervention arms: InTandem or a treatment-matched Active Control. Randomization was performed through the Electronic Data Capture (EDC) system (Castor, New York, New York) without stratification using block randomization with randomly selected block sizes to ensure approximately equal group sizes. Trial investigators and participants were not blinded to randomization. InTandem was supplied to trial sites by MedRhythms, Inc. as the MR-001 neurorehabilitation system (Portland, ME).

### Trial design overview

This randomized controlled trial was conducted in clinical settings under the supervision of clinical investigators to evaluate safety and efficacy relative to a treatment-matched Active Control group. Because InTandem is designed for independent use in the home, a separate study has been designed to evaluate its independent set-up and operation[34].

Regardless of intervention arm assignment, all participants were scheduled to complete 15 intervention sessions over a 5-week period, at a frequency of 3 sessions per week, and with each session consisting of 30 min of walking. A complete session was defined as 15-to-30 minutes of walking. Participants were not allowed to use assistive devices during the intervention session but were allowed to take rest breaks as needed or when investigators believed that a rest break was necessary. Because sessions ended automatically after 30 min, rest breaks counted towards each intervention session (i.e., a 5-minute rest break resulted in 25 min for intervention).

All intervention sessions occurred on an overground track or hallway of at least 100 feet in length. In addition, the space used for intervention sessions could not have any discernible background music or audible distractions that could compete with the auditory intervention. Though no cueing or walking instructions were provided to study participants during training, all study participants were provided general instructions at the start of each intervention session (see Supplementary Information for the language recommended in the clinical trial protocol). During each intervention session, physiological parameters (i.e., heart rate, blood pressure, and/or respiration rate) were monitored according to each site's standard procedures, or at an investigator's discretion to ensure safety of each study participant. If a study participant's measurements fell outside of the acceptable parameters set by the site or the study participant's physician, the intervention session could be paused or terminated based on a collaborative decision made by the study participant and investigator.

If required, participants were allowed to restart their intervention schedule after a washout period ($n = 6$). The minimum washout period required was equal to the duration of intervention weeks already completed. Following the washout period, trial eligibility was re-evaluated. Only trial data collected after the washout period were used in the trial's primary analyses.

### InTandem system and intervention

InTandem consists of a touchscreen device locked with a preloaded proprietary software application, a headset for delivery of the rhythmic auditory stimulus, two shoe-worn inertial sensors, and charging equipment (Fig. 3). To produce the individualized and progressive intervention, personalized audio cues are embedded into time-shifted music based on real-time decisions made by closed-loop control algorithms that continuously assess the gait data collected by the inertial sensors.

InTandem's closed-loop control of music includes two real-time algorithm components that operate in parallel. The first component assesses the user's ability to entrain to the target tempo based on real-time measurements of step-to-beat alignment, defined as the ratio of the user's walking cadence (i.e., steps per minute) to the tempo of the time-shifted music. Here "time-shifted" refers to both: (1) stabilization of the natural variability (if any) of the song's original tempo and (2) the within-session adaptive tempo increases or decreases used to individualize the intervention. More specifically, the initial target tempo is set to the user's baseline cadence, as measured by the inertial sensors during an un-cued baseline walk, with subsequent modulation of the target tempo based on assessments made by the second algorithm component during the intervention. The second component assesses the user's gait symmetry and variability based on real-time sensor input during the intervention, with symmetry measured as a ratio of interlimb stance and swing times, and variability measured as the coefficient of variation of stride times.

The entrainment and gait quality values from the two components are then compared to proprietary thresholds, which the algorithm uses to make two decisions. The first decision made is if it's appropriate to modulate the song tempo—doing so requires concurrence by both components that (1) the user is sufficiently entrained and (2) their gait quality is appropriate for progression; only if both criteria are achieved is it considered safe to increase the tempo. The specific criteria used to make these determinations are based on the developer's clinical experience delivering rhythmic auditory stimulation interventions, which were further refined during development to accommodate walking in real-world settings. The second decision made is if the user requires the addition of a synchronized rhythm track to the music to enhance the beat salience. When the rhythm track is introduced, the volume of the music adjusts so that the user hears more of the rhythm track relative to the music. Once the user is entraining and their gait symmetry and variability is within acceptable ranges, the rhythm track is then removed. If a user does not entrain, even with the added rhythm track, the music of the tempo will decrease until the user is entraining. The cascade of decisions that tailor the intervention to a person's gait on a given day allows for personalized treatment with every session.

The music used in the trial was screened to ensure therapeutic suitability. This process ensured that the music met requirements for beat prominence and tempo. More specifically, the music included with the InTandem system is a core part of the intervention; not every piece of music is fit for purpose. Using a proprietary screening process, we assess high and low level features of the audio content, such as average tempo (what cadences could this song work for?), time signature (is the song in duple or triple meter?), duration of song (is it too long or short of a song?), and beat strength (how prominent is the beat?), in addition to other feature analyses. Through this process, a song is either accepted or rejected for use in InTandem. For this trial, the song list included a variety of genres, such as rock and roll, pop, and disco, and songs were screened and compiled as master recordings from recording music artists from a music licensing platform (Songtradr, Los Angeles, California).

### Active control

The Active Control intervention was matched to the InTandem intervention in number, duration, and frequency of sessions. Like InTandem sessions, Active Control sessions consisted of 30 min of overground walking practice supervised by clinical investigators. Though Active Control participants did not wear the InTandem system, they did wear inertial sensors on their shoes to enable recording gait data during each session.

### COVID-19 considerations

The trial was disrupted by the COVID-19 PHE. In brief, an unplanned interim analysis was conducted during the research shutdown,

requiring revision of the statistical plan. More specifically, alpha was adjusted from 0.05 to 0.025. Moreover, according to guidance from FDA, the impact of COVID-19 on the trial data was examined by assessing the comparability of participants enrolled before versus during the COVID-19 PHE. The investigation revealed an effect of COVID-19 that was reasonably explained by a non-random change in trial management at one center, and ultimately required administrative removal of 8 individuals to resolve the observed effect of COVID-19. In brief, eight individuals were found to have been recruited into the trial during the COVID-19 PHE shortly after first completing another walking intervention trial. The washout period imposed between the two competing trials was discovered to be inconsistent with the precedent set in the InTandem trial protocol for a washout period. This deviated recruitment strategy during the COVID-19 PHE was considered an unanticipated event that affected only these eight participants; their administrative removal resulted in COVID-19 no longer affecting the trial data.

### Statistical analyses

Data analyses were completed using IBM SPSS Statistics (Version 27) software. Safety was described as the frequency, severity, and relatedness of treatment-emergent AEs. An AE was defined as any untoward or unfavorable physical or psychological occurrence and included any abnormal sign, symptom, or disease temporally associated with participation in the research. Descriptive statistics are reported using means and standard deviations (SD) for continuous variables and proportions for categorical variables, unless noted otherwise. A $2 \times 2$ General Linear Mixed Model (GLMM) was used to evaluate differences in the treatment effect between groups (i.e., a Treatment x Time interaction), which allowed for nesting participants within Centers. Effects of interest in the model included Treatment (InTandem vs. Active Control), Time (Pre-intervention vs. Post-intervention), and the Treatment x Time interaction. Box-Cox and Shapiro-Wilk tests were used to evaluate linearity and normality. Outliers and influential cases were screened for and removed[36]. Robust standard errors were used to account for violations of normality. A compound symmetric covariance matrix was used to model the correlation structure among residuals and Satterthwaite Approximation was used to adjust the degrees of freedom for significance testing.

For the study primary endpoint of a change in gait speed overtime between groups, this study was originally powered (power = 0.80) to detect a moderately large to large effect ($d = 0.64–0.82$) for a plausible range of Intraclass correlation coefficients (ICCs), ($r = 0.05$ to $r = 0.20$), and conservative estimates of the correlation among repeated measures ($r = 0.4$ to $r = 0.6$) for $n = 66$ and alpha = 0.05.

Between-group differences in the number of responders were evaluated using Chi-Square ($\chi^2$) tests. To account for an unplanned interim trial analysis during the COVID-19 Public Health Emergency (PHE), alpha was lowered to 0.025. Responders were defined using two criteria: (1) a post-intervention change in 10 mWT speed that surpassed the 0.16 m/s MCID[31] or (2) *both* a post-intervention change larger than 0.16 m/s and a final post-intervention speed above 0.80 m/s[32].

In addition to the primary endpoint analyses, an exploratory time-course of change analysis was conducted using a $2 \times 17 \times 2$ GLMM. This analysis tested between-group differences in the treatment effect using all 10 mWT speed data collected. The model tested three main effects: Treatment (InTandem vs. Active Control), Time (Baseline, Session 1, …, Session 15, and Closing), and Trial (pre- vs. post-session walking speed), and their two-way interactions. Alpha was set to 0.05 for this analysis.

### Reporting summary

Further information on research design is available in the Nature Portfolio Reporting Summary linked to this article.

## Data availability

The individual participant data that underlie the findings of this study are included in the figures and in a Supplementary Information file. All study data are available on request. Individual participant data, after deidentification, (including data dictionaries) will be shared. Proposals for additional data analyses should be directed to staylor@medrhythms.com. All proposals will be reviewed to ensure that they are complete and valid, and that the data are available, consistent with participant privacy and informed consent. Responses to data requests will be provided within two months. To gain access, data requestors will need to sign a data access agreement and provide a methodologically sound proposal. The study protocol is available on the trial registration page (https://clinicaltrials.gov/study/NCT04121754) and is also available on request. Source data have been provided with this paper and its supplementary information files. Source data are provided in this paper.

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

## Acknowledgements

We acknowledge contributions by the following study staff: at BU: Johanna Spangler, Lillian Braga-Ribeirinha, Joan Breen, Terry Ellis; at SR: Sara Prokup, Kristine Buchler, Matt Gifforn, Kelly McKenzie, Matt McGuire, Jen Traines; at KF: Melvin Mejia, Brandon Ross, Corey Greene, Kathleen Chervin, Sharon Franco, Greg Ames, Oluwaseun Ibironke; at UNC-Chapel Hill: Chelsea Duppen, Alex Huntsinger, David Rowland; at SRH: Catherine Adans-Dester, Anne O'Brien; at AH: Jillian Cummings; at JH: Kendra M. Cherry-Allen, Margaret A. French, Spencer Gonzaga. We also acknowledge the following employees at MedRhythms, Inc.: Kirsten Smayda, Ph.D. for her manuscript editing support; Jennifer Lavanture and Owen McCarthy for their work during protocol development; and Brian Bousquet-Smith, Nicholas LaJoie, and Chrissy Stack for their Clinical Operations support during the conduct of the trial.

## Author contributions

L.N.A. contributed first and subsequent drafts, visualization, and secondary analyses. A.J. contributed a critical review. K.J.N. contributed a critical review. M.D.L. contributed critical review and framing strategy. P.B. contributed a critical review. M.N. contributed critical review. D.P. contributed a critical review. P.R. contributed a critical review. R.T.P. contributed statistical analyses, data planning, and visualizations. B.A.H. contributed a critical review. D.A.P. contributed critical review. S.R.T. contributed a critical review.

## Competing interests

This trial was funded by MedRhythms, Inc. L.N.A. is a paid advisor to MedRhythms Inc. B.A.H. is co-founder and CEO of MedRhythms Inc. with equity interest. S.R.T. and D.A.P. are employees of MedRhythms Inc. with equity interest. This study was completed under a master management plan instituted by the Boston University Financial Conflicts of Interest Committee. The remaining authors declare no competing interests.
