## [Peer Review File · Nature Communications]

Efficacy and safety of using auditory-motor entrainment to improve walking after stroke: A multi-site randomized controlled trial of InTandemTMEditorial Note: This manuscript has been previously reviewed at another journal that is not operating a transparent peer review scheme. This document only contains reviewer comments and rebuttal letters for versions considered at *Nature Communications*. Mentions of prior referee reports have been redacted.

REVIEWERS' COMMENTS

Reviewer #1 (Remarks to the Author):

[REDACTED]. The study demonstrates an impressive result showing clinical and statistical change (vs active control) in walking speed within a relatively short period of time. The active control was robust, lending validity to the findings. There is a clear need to improve walking speed post-stroke, so there is also an important clinical utility for such an intervention.

[REDACTED]. I have a few suggestions that I believe would improve the paper.

The addition of the additional data on gait symmetry and participants' responses would be a major improvement. I see the authors' point that it is cumbersome to report these additional findings in the primary effects paper – as not all people completed the assessments, etc, - but without those results the reader is left hanging – particularly regarding the gait symmetry change, as it is emphasized as a key factor in the intervention. To not report on it leaves a major hole.

[REDACTED]. If word count allows I would suggest adding some rationale to various pieces of the methods or results. [REDACTED]. It is a known problem with responder analyses in gait studies and I'm not sure I've seen someone explicitly state the rationale as outlined here on this problem (picking one threshold vs another in a responder analysis). Further, I think the suggestion of future studies adopting this approach would be appropriate to include here.

Another example of where more rationale could be included is the choice to not do a follow-up assessment. Could that be included the methods? It is a notable absence, thus stating it in the methods before the results would help to set the stage for the results.

[REDACTED].

Minor:

What type of music was used? The authors provide more information about tempo etc, but did participants get to select a genre? Ie, Rock and roll, country, blues? Are they known popular songs or is it generated music? Can you provide an example of the songs

Reviewer #2 (Remarks to the Author):

[REDACTED]. I have a few minor concerns that remain.

1) [REDACTED]. The authors may consider making brief statements in the methods and/or discussion about a) the capabilities of the InTandem device but that the current paper focuses on gait speed for the purposes of evaluating efficacy and safety, and b) future work will investigate the additional parameters measured

2) [REDACTED]. It is interesting to note that in the active control group stance time asymmetry increases within a session suggesting that improving speed may come at a cost of poor gait quality which further supports the rationale for the design of InTandem.

Efficacy and safety of using auditory-motor entrainment to improve walking after stroke: A multi-site randomized controlled trial of InTandem™

Response to Reviewers

Dear Colleagues,

Thank you for the careful second review of our manuscript. In this document, we provide a point-by-point response to each of the remaining comments and suggestions. To facilitate your review, where helpful, the modified sections of the manuscript have been included with revisions shown using **boldface** font.

Reviewer #1:

Comment 1: The addition of the additional data on gait symmetry and participants' responses would be a major improvement. I see the authors' point that it is cumbersome to report these additional findings in the primary effects paper – as not all people completed the assessments, etc, - but without those results the reader is left hanging – particularly regarding the gait symmetry change, as it is emphasized as a key factor in the intervention. To not report on it leaves a major hole.

Response and Action: We thank the reviewer for their feedback. Due to the apparent contradiction between the Reviewers on the need to include these secondary analyses, and in accordance with the journal recommendation to not refer to ongoing work and unpublished data in the manuscript, we now clearly state in the *Limitations* section of the *Discussion* the absence of these data and that further study is warranted in order to address these comments. The text additions are as follows:

...the trial's primary efficacy analysis did not include other important outcome measures, such as gait biomechanics, patient-perceived benefit, self-efficacy, and community walking activity; the full extent to which post-stroke walking can be improved by InTandem is thus not known.

Future clinical trials are warranted to assess InTandem's generalizability to other subgroups and patient populations, as well as any additional benefit from longer treatment periods, durability of treatment effect, and impact on other important outcome measures, including gait biomechanics. The evaluation of InTandem's effects on post-stroke gait, within and across intervention sessions, is a natural area for future study given that the InTandem system inherently measures gait parameters to individualize and progress the auditory-motor intervention.

Comment 2: [REDACTED]. If word count allows I would suggest adding some rationale to various pieces of the methods or results. [REDACTED]. It is a known problem with responder analyses in gait studies and I'm not sure I've seen someone explicitly state the rationale as outlined here on this problem (picking one threshold vs another in a responder analysis). Further, I think the suggestion of future studies adopting this approach would be appropriate to include here.

Response and Action: Thank you for the opportunity and suggestion to more explicitly state our rationale in the manuscript, and use this rationale to suggest future studies adopt this powerful approach. We have added the following text from our response document to the second paragraph of the *Discussion*. We believe placing this rationale in the *Discussion* helps place the findings in stronger context, and this positioning in the manuscript also allows more natural suggestion for future studies adopt this approach.

Critically, regardless of the responder analysis used, InTandem participants were three times more likely to be responders. The a priori selection of two responder analyses for this study was based on extensive discussions with different stakeholders (i.e., FDA, users, prescribers, and payers), wherein different groups were found to value the two responder analyses differently. In brief, the most common approach to defining a responder is to use an MCID cutoff; however, a key limitation of this approach is that a subject's walking speed change may surpass the MCID but not be sufficient to place their absolute walking speed above clinically meaningful thresholds (e.g., 0.80

m/s is a walking speed threshold that is thought to be the minimum required for community walking). In contrast, a key limitation to defining a responder only as someone who surpasses an absolute walking speed threshold, without regard for the magnitude of their change in speed, is that a modest, non-clinically important change could be sufficient (e.g., a 0.02 m/s change for someone with a baseline speed of 0.79 m/s would raise their absolute walking speed over the 0.80 m/s threshold). For our study, by having the first responder analysis focus on the 0.16 m/s MCID cutoff, the analysis is able to compare well to other papers in the field. And by having our second responder analysis combine both criteria (i.e., a change > 0.16m/s and a post-training speed of > 0.80 m/s), we are able to address the limitations to using each alone. Ultimately, though these two responder analyses produced similar results in our study, this may not be true for other intervention studies, and we would encourage others to consider this dual approach in future designs to maximize the scientific reach of their work.

Comment 3: Another example of where more rationale could be included is the choice to not do a follow-up assessment. Could that be included in the methods? It is a notable absence, thus stating it in the methods before the results would help to set the stage for the results.

Response and Action: Thank you for this excellent suggestion. Due to journal formatting guidelines, which require the *Results* and *Discussion* sections to come before the *Methods*, the rationale for not including a dedicated follow-up timepoint that we had included originally in the *Limitations* subsection of the *Discussion*, now comes before the *Methods*.

Comment 4: [REDACTED].

Response and Action: Thank you for this suggestion. We have made this change as suggested:

*“More specifically, InTandem resulted in a 27% larger increase in 10mWT speed (i.e., 0.14 m/s vs. 0.11 m/s) in 37% less time (i.e., 5 weeks vs. 8 weeks). However, it should be noted that Boyne et al report continued gains in walking speed, up to 0.19 m/s, after 12 weeks of high-intensity training, **which we are unable to directly compare against given the shorter intervention duration of our study.** Nonetheless, the trajectory of improvement observed in our time-course of change analysis (Fig 2C) may suggest InTandem’s potential to produce larger gains than high-intensity interval training in less time.*

Comment 5: What type of music was used? The authors provide more information about tempo etc, but did participants get to select a genre? I.e, Rock and roll, country, blues? Are they known popular songs or is it generated music? Can you provide an example of the songs

Response and Action: Thank you for your comment. We have made the following clarifying additions to the *Methods* section of the manuscript:

*“The music used in the trial was screened to ensure therapeutic suitability. This process ensured that the music met requirements for beat prominence and tempo. More specifically, the music included with the InTandem system is a core part of the intervention; not every piece of music is fit for purpose. Using a proprietary screening process, we assess high and low level features of the audio content, such as average tempo (what cadences could this song work for?), time signature (is the song in duple or triple meter?), duration of song (is it too long or short of a song?), and beat strength (how prominent is the beat?), in addition to other feature analyses. Through this process, a song is either accepted or rejected for use in InTandem. **For this trial, the song list included a variety of genres such as rock and roll, pop, and disco, and songs were screened and compiled as master recordings from recording music artists.**”*

Reviewer #2:

Comment 1: [REDACTED]. The authors may consider making brief statements in the methods and/or discussion about a) the capabilities of the InTandem device but that the current paper focuses on gait speed for the purposes of evaluating efficacy and safety, and b) future work will investigate the additional parameters measured.

1A Response and Action: We thank the reviewer for their feedback. Due to the apparent contradiction between the Reviewers on the need to include these secondary analyses, and in accordance with the journal recommendation to not refer to ongoing work and unpublished data in the manuscript, we now clearly state in the *Limitations* section of the *Discussion* the absence of these data and that further study is warranted in order to address these comments. See *Response and Action to Reviewer 1's Comment 1*.

Comment 2: [REDACTED]. It is interesting to note that in the active control group stance time asymmetry increases within a session suggesting that improving speed may come at a cost of poor gait quality which further supports the rationale for the design of InTandem.

Response: Thank you. We agree and look forward to sharing these data in a complete follow-up manuscript.